# Clinical Safety and Effectiveness of Robotic-Assisted Surgery in Patients with Rectal Cancer: Real-World Experience over 8 Years of Multiple Institutions with High-Volume Robotic-Assisted Surgery

**DOI:** 10.3390/cancers14174175

**Published:** 2022-08-29

**Authors:** Ching-Wen Huang, Po-Li Wei, Chien-Chih Chen, Li-Jen Kuo, Jaw-Yuan Wang

**Affiliations:** 1Division of Colorectal Surgery, Department of Surgery, Kaohsiung Medical University Hospital, Kaohsiung Medical University, Kaohsiung 80756, Taiwan; 2Department of Surgery, Faculty of Medicine, College of Medicine, Kaohsiung Medical University, Kaohsiung 80756, Taiwan; 3Division of Colorectal Surgery, Department of Surgery, Taipei Medical University Hospital, Taipei Medical University, Taipei 110301, Taiwan; 4Department of Surgery, School of Medicine, College of Medicine, Taipei Medical University, Taipei 110301, Taiwan; 5Department of Surgery, Koo Foundation Sun Yat-Sen Cancer Center, Taipei 112019, Taiwan; 6Graduate Institute of Clinical Medicine, College of Medicine, Kaohsiung Medical University, Kaohsiung 80756, Taiwan; 7Graduate Institute of Medicine, College of Medicine, Kaohsiung Medical University, Kaohsiung 80756, Taiwan; 8Center for Cancer Research, Kaohsiung Medical University, Kaohsiung 80756, Taiwan; 9Pingtung Hospital, Ministry of Health and Welfar, Pingtung 900214, Taiwan

**Keywords:** clinical safety and effectiveness, robotic-assisted rectal surgery, high-volume, real-world evidence, multi-institutional study

## Abstract

**Simple Summary:**

The aim of this retrospective observational study was to evaluate perioperative and short-term oncological outcomes of robotic-assisted rectal surgery (RRS) in hospitals with a high-volume of robotic-assisted surgeries. This study enrolled patients with rectal adenocarcinoma undergoing RRS from three high-volume institutions from December 2011 to June 2020. Compared with other studies, our results revealed the equivalent or superior perioperative and short-term oncological outcomes. Hence, RRS is an effective, safe, and feasible technique for patients with rectal cancers in high-volume hospitals.

**Abstract:**

The perioperative and short-term oncological outcomes of robotic-assisted rectal surgery (RRS) are unclear. This retrospective observational study enrolled patients with rectal adenocarcinoma undergoing RRS from three high-volume institutions in Taiwan. Of the 605 enrolled patients, 301 (49.75%), 176 (29.09%), and 116 (19.17%) had lower, middle, and upper rectal cancers, respectively. Low anterior resection (377, 62.31%) was the most frequent surgical procedure. Intraoperative blood transfusion was performed in 10 patients (2%). The surgery was converted to an open one for one patient (0.2%), and ten (1.7%) patients underwent reoperation. The overall complication rate was 14.5%, including 3% from anastomosis leakage. No deaths occurred during surgery and within 30 days postoperatively. The positive rates of distal resection margin and circumferential resection margin were observed in 21 (3.5%) and 30 (5.0%) patients, respectively. The 5-year overall and disease-free survival rates for patients with stage I–III rectal cancer were 91.1% and 86.3%, respectively. This is the first multi-institutional study in Taiwan with 605 patients from three high-volume hospitals. The overall surgical and oncological outcomes were equivalent or superior to those estimated in other studies. Hence, RRS is an effective and safe technique for rectal resection in high-volume hospitals.

## 1. Introduction

Colorectal cancer (CRC) is the third most common malignancy type and the third leading cause of cancer-related mortality worldwide [1]. In 2017, approximately 1.8 million new CRC diagnoses and 896,000 CRC-related mortalities were reported worldwide [2]. Since 2006, CRC has been the most common cancer type, and its prevalence has increased rapidly in Taiwan. In 2006 and 2018, the incidences were 45.5 and 66.3 per 100,000, respectively (with 10,398 and 16,525 new diagnoses, respectively) [3]. Moreover, CRC is the third leading cause of cancer-related mortality. In 2020, 6489 people in Taiwan died of CRC, with the mortality rate being 27.5 and 21.2 per 100,000 individuals in 2020 and 2010, respectively [3].

In the past decades, the improved treatment outcomes of rectal cancers have been due largely to several factors, including novel therapeutic modalities and improved surgical approaches. Preoperative concurrent chemoradiotherapy (CCRT) has been reported to be beneficial for patients with locally advanced rectal cancer (LARC) [4,5,6]. Therefore, preoperative CCRT is the standard treatment for patients with LARC. Moreover, several perioperative benefits have been reported with laparoscopic rectal surgery, including low postoperative pain, early mobilization, early postoperative recovery, and a short hospital length of stay (LOS) [7,8,9]. Technically skilled surgeons experienced in laparoscopic rectal surgery are needed for the patient to gain perioperative benefits because it is difficult to perform laparoscopic rectal surgery within the narrow space of the pelvis using rigid laparoscopic instruments that inherently have limited dexterity and range of motion. The robotic surgical system provides numerous advantages, such as high-definition three-dimensional vision with up to 10× magnification, stable traction by robot arms, and the availability of articulatory instruments and a surgeon-controlled camera platform. Compared with open surgical and conventional laparoscopic approaches for patients with rectal cancers, robotic-assisted rectal surgery (RRS) appears to be favorable in terms of perioperative and short-term oncological outcomes [10,11,12,13,14,15]. Studies on RRS in Taiwan have demonstrated that RRS is safe and feasible for high dissection and low or selective ligation of the inferior mesentery artery, for the single-docking technique, in cases of long intervals between the completion of radiotherapy and robotic-assisted surgery, and for older adult patients aged >70 years [16,17,18,19,20,21,22]. However, these studies have been conducted in a single institution and have had small sample sizes. Therefore, we conducted a retrospective study in Taiwan covering multiple institutions using empirical data pertaining to high-volume robotic-assisted surgery.

## 2. Materials and Methods

### 2.1. Patients

This retrospective observational study enrolled patients with rectal cancer undergoing robotic-assisted surgery from four surgeons at three high-volume institutions in Taiwan, namely Kaohsiung Medical University Hospital, Taipei Medical University Hospital, and Koo Foundation Sun Yat-Sen Cancer Center, at any period from December 2011 to June 2020. The surgeons were new to robotic-assisted surgery in 2011. They contributed 211, 145, 127 and 118 cases, respectively. The inclusion criteria were (1) histologically confirmed rectal adenocarcinoma with the tumor located within 15 cm from the anal verge and (2) absence of second primary cancer. The exclusion criteria were (1) having received emergent surgeries or (2) being lost to follow-up after robotic-assisted surgery. In total, 605 eligible patients received robotic-assisted surgery with the da Vinci Si or Xi surgical system (Intuitive Surgical, Inc., Sunnyvale, CA, USA). This study was approved by the Institutional Review Boards (IRBs) of Kaohsiung Medical University Hospital, Taipei Medical University Hospital, and Koo Foundation Sun Yat-Sen Cancer Center (KMUHIRB-E(I)-20200036, N202102060, N202103023, 20210304A, respectively).

Preoperative staging studies included a colonoscopy and computed tomography or high-definition magnetic resonance imaging of the abdomen or pelvis in all patients. On the basis of the distance from the anal verge, rectal cancer was categorized into upper (11–15 cm), middle (6–10 cm), and lower (≤5 cm) rectal cancer. Patients with LARC (i.e., T3, T4, or N+ rectal cancer) underwent preoperative CCRT, which was (1) 5-fluorouracil-based chemotherapy or a FOLFOX (i.e., 5-fluorouracil, leucovorin, and oxaliplatin) regimen every 2 weeks with radiotherapy (long course or short course), (2) chemotherapy only, or (3) radiotherapy (short course) only. Furthermore, patients with cT2 rectal cancer located within 5 cm from the anal verge underwent the same preoperative treatment for sphincter preservation.

The following clinicopathological features and perioperative parameters were evaluated: age, sex, TNM (tumor, node, and metastasis) classification, tumor location (distance from the anal verge and categorized as lower third, middle third, upper third, and unknown), body mass index (BMI), American Society of Anesthesiologists (ASA) score, and Charlson Comorbidity Index (CCI) score. The TNM classification was determined according to the criteria of the American Joint Commission on Cancer (AJCC) and International Union Against Cancer (UICC) [23]. Intraoperative safety measures pertained to events that occurred during the surgery, specifically death during surgery, surgical procedures [16,17,24,25,26], and conversion to open surgery, and various measures, specifically docking time, operation time, console time, estimated blood loss, and blood transfusion, were collected. Postoperative clinical outcomes were analyzed for predischarge and postdischarge periods, including LOS, rehospitalization within the 30-day postoperative period, reoperation within the 30-day postoperative period, and death within the 30-day postoperative period.

### 2.2. Data Management

#### Confidentiality and Quality Control

Patient baseline information and clinical outcomes recorded in medical charts and operative notes were reviewed and retrieved retrospectively. All data were collected and recorded in a standardized case report form format by an investigator affiliated with each hospital, and then each dataset was pooled to create a multi-institutional dataset. The research data were stored in a password-protected database kept in an external hard drive under the care of the principal investigator (PI) and were accessible only by researchers. All investigators complied with the Personal Data Protection Act. Patient data were de-identified and a pseudo code was assigned to each patient to protect their identity. The key investigator and the lead researcher of each hospital reviewed the data entered to ensure that the data were accurate.

### 2.3. Study Monitoring and Ethical Consideration

#### 2.3.1. Monitoring and Inspecting

The PI of each hospital allocated adequate time for monitoring activities and ensured that the supervisor or other compliance or quality assurance reviewer was given access to all study-related documents (e.g., source documents, datasets, collected data). Participation as an investigator in this study implied acceptance of potential inspection by government regulators and applicable hospital compliance and quality assurance officers.

#### 2.3.2. Ethical Consideration

This study was conducted in accordance with Taiwanese regulations and research ethics policies and procedures of the authors’ institutions. The PI was responsible for informing the IRB and research groups regarding any amendments to the protocol or study-related documents.

### 2.4. Statistical Analysis

Descriptive statistics were used to analyze patient characteristics and the outcomes of robotic-assisted surgery. Continuous variables were summarized using the mean, standard deviation (SD), median, and 25 and 75 percentiles (IQR, interquartile range), whereas categorical variables were summarized using frequencies and percentages (%). All data were statistically analyzed using the Excel software (Microsoft Corporation, Inc., Redmond, WA, USA) and R software (Free Software Foundation, Inc., Boston, MA, USA). All patients were followed up regularly until their death or their last follow-up date, whichever occurred first. The console time was defined as the total duration of robotic-assisted surgical procedures with the robotic system (da Vinci Si or Xi surgical system, Intuitive Surgical, Inc., Sunnyvale, CA, USA). The operation time was defined as the total duration between the initial skin incision and wound closure completion. Disease-free survival (DFS) was defined as the duration between the date of primary treatment and the date of diagnosis of recurrence or metastatic disease or last follow-up. The overall survival (OS) time was defined as the duration between the date of primary treatment and the date of all-cause death or last follow-up. The Kaplan–Meier method was used to evaluate DFS and OS, and a log-rank test was performed to compare time-to-event distributions. A *p* value of <0.05 indicated statistical significance.

## 3. Results

### 3.1. Patient Characteristics and Perioperative Outcomes

Between December 2011 and June 2020, 605 patients with rectal cancer undergoing RRS at three high-volume institutions in Taiwan were enrolled. The demographic and baseline characteristics of the patients are summarized in Table 1. The median age of patients was 60 years (IQR, 51–67 years). Moreover, 301 (49.75%), 176 (29.09%), and 116 (19.17%) patients had lower, middle, and upper rectal cancers, respectively; the tumor location of 12 (1.98%) patients were unknown. Preoperative treatment was administered to 454 patients (75%), including CCRT, chemotherapy, and radiation to 429 (70.8%), 7 (1.2%), and 18 (3.0%) patients, respectively. In total, 536 (88.6%), 28 (4.63%), and 41 (6.78%) patients had a CCI score of 0–1, 2, and ≥3, respectively. Furthermore, 21 (3.49%), 422 (70.1%), 157 (26.08%), and 2 (0.33%) patients had ASA scores I, II, III, and IV, respectively. The most frequent surgical procedure was low anterior resection (LAR) (377, 62.3%), followed by intersphenteric resection (ISR) with coloanal anastomosis (200, 33.1%), and abdominoperineal resection (APR; 28, 4.6%).

### 3.2. Intraoperative Safety and Clinical Outcomes

Table 2 summarizes the intraoperative safety and perioperative outcomes of the patients. The median console time and operating time were 211 (IQR, 172–256) and 270 (IQR, 210–335) minutes, respectively. The historical trend of operation time significantly decreased during this study (*p* < 0.001, Figure 1A). The median estimated blood loss was 50 mL (IQR, 30–100 mL). Only one (0.2%) patient required conversion to open surgery. Moreover, ten (1.7%) patients underwent reoperation within the 30-day postoperative period and the causes of reoperation were surgical site infection (6 patients), ileus (3 patients), and anastomotic leakage (one patient). No deaths occurred during the surgery and within the 30-day postoperative period. The mean length of postoperative LOS was 13.51 days (SD = 7.93), which decreased with year (Figure 1B, Menn–Kendall, *p* = 0.002).

### 3.3. Pathological Outcomes and Oncological Outcomes

The pathological outcomes of all 605 patients are listed in Table 2. The median number of harvested lymph nodes was 14 (IQR, 10–20). The distal resection margin (DRM) and circumferential resection margin (CRM) were positive in 21 (3.5%) and 30 (5.0%) patients, respectively.

The median follow-up duration of the 605 patients from the primary treatment was 47.1 (range, 1.7–110.3) months. Among the 605 patients, local recurrence and distant metastases were noted in 18 (3.0%) and 95 (15.7%) patients, respectively. At a median follow-up duration of 47.1 months, the 5-year OS was 91.1% and 5-year DFS was 86.3% (Figure 2) for patients with stage I–III rectal cancer.

### 3.4. Postoperative Complications

The postoperative complications are summarized in Table 3. The overall complication rate was 14.4% (87/605). The most common postoperative complications were infection events and ileus. Infection events, including intraabdominal infection, intraabdominal abscess, and surgical site infection, were observed in 22 (3.6%) patients. Ileus, anastomosis leakage, and urinary retention were observed in 20 (3.3%), 18 (3.0%), and 6 (1.0%) patients, respectively. According to the Clavien–Dindo Classification, 88.5% (77/87) of postoperative complications were of grade I, and 11.5% (10/87) were of grade III. The patients with grade I complications recovered uneventfully after conservative treatment.

## 4. Discussion

In the present study, we collected the demographic, baseline, perioperative, and post-operative data of 605 patients with rectal adenocarcinoma undergoing RRS from three high-volume institutions in Taiwan between December 2011 and June 2020. To the best of our knowledge, this is the first study with real-world data from multiple institutes. Furthermore, this study has the largest RRS data collection in Taiwan with the longest follow-up. We believe our data are representative of the status of RRS and its safety and clinical outcomes in Taiwan. Nevertheless, because this study adopted a single-arm design, we compared our results with those of the literature to assess safety and clinical efficacy of RRS in Taiwan.

### 4.1. Baseline Characteristics

The patients’ baseline characteristics of our study were comparable with those in the literature. The risks of CRC and predictive mortality were positively associated with age [27]. In our study, the patients’ median age was 60 (IQR, 51–67), similar to that in the literature, and patients with age ≥70 accounted for 19.17% of the sample. The female percentage (43%) of our study was slightly higher than that in the literature (32–37.6%) [28,29,30].

As for baseline patient health status, 11.4% of the patients had CCI ≥ 2, which was slightly higher than that of previous studies (of 4.9%) [31]. The ASA score reflects patient comorbidity before the surgery, and a score of ≥3 constitutes an independent risk factor for postoperative complications [32]. In the present study, 26.4% of patients had an ASA score of ≥3, which was much higher than estimates in the literature (0–11.7%). Preoperative CCRT requisites and regimen varies depending on country. In the present study, 70.8% of our patients received preoperative CCRT, which was higher than the estimates in the literature (3.5–46.8%) [29,33], but comparable with that reported in a Korean RCT (77.3%) [34].

### 4.2. Operation Time

All centers estimated operation time from skin to skin. The mean operation time in our study was 284 (SD: 101) minutes, which was comparable with previously reported outcomes in the literature (Table 4). The most recent systematic literature review of eight RCTs reported that the pooled operation time of RRS was 23 min longer than that of laparoscopic surgery (*p* = 0.019) [35]. Operation time is considered an indicator of how much the surgeon is on the other side of the learning curve. In our study, operation time decreased during the study duration, which indicates an improvement in surgeons’ skills and efficiency as they work on more cases. The steepest improvement in operating time is known to occur in the initial 15–40 cases, but our study found that even when surgeons supposedly plateaued in their skill, the operation time consistently decreased from 280 to 240 min.

### 4.3. Conversion

Conversion to open surgery during minimally invasive surgery is known to be one of the prognostic factors that lead to an increased LOS, a high complication rate, and a high cancer recurrence rate [24,40]. In our study, the conversion rate was estimated to be 0.2%, which was much lower than that reported in the published literature. A previous systematic literature review indicated that the median conversion rate of a laparoscopic group was 10%, with a range of 6.4–57.6% [41]. Furthermore, the most recent systematic literature review of eight RCTs observed that the pooled conversion rate was significantly lower in the RRS group (5.72%) than in the laparoscopic surgery group (11.89%; OR = 2.215, 95% CI = 1.357–3.6315, *p* = 0.001) [35]. Furthermore, a similar trend was observed in the ROLARR study, which was an RCT comparing 471 patients who underwent RRS or laparoscopic surgery across ten countries [42]. In the ROLARR study, 19 of 236 patients (8.1%) in the RRS group and 28 of 30 patients (12.2%) in the laparoscopic group had their surgery converted to an open one. A recent study that investigated 50,855 patients using the US National Cancer Database noted that the conversion rate of RRS was significantly lower than that of laparoscopic surgery (RRS: 7.0%, laparoscopic surgery: 15.7%, *p* < 0.0001) [43]. The relatively lower conversion rate may be attributed to high-volume robotic-assisted surgeons in the present study or because of relatively lower BMI compared to Western studies.

### 4.4. Circumferential Resection Margin Positivity

The CRM is the closest margin between the deepest penetration of the tumor and the edge of resected soft tissue around the rectum or from the edge of a lymph node. In the present study, we investigated cancer positivity in CRM, and 4.96% of patients were tested positive. Furthermore, the reported CRM was lower than that previously reported in the literature. In the ROLARR study, 5.1% and 6.3% of patients exhibited positive CRM in RRS and laparoscopic groups (*p* = 0.56) [40]. No statistically significant difference was observed between these two groups. In the COLOR II study, Positive CRM was noted in 10% of both the laparoscopic and open surgery groups (*p* = 0.850) [7].

### 4.5. Harvested Lymph Node

The number of lymph nodes examined after surgery and the assessment of tumor metastasis to regional lymph nodes are key to an accurate diagnosis of cancer staging. The AJCC/UICC recommends for at least 12 lymph nodes to be examined for each surgical specimen of CRC [44]. Another study suggested that OS improves with the numbers of lymph nodes retrieved [45]. In the present study, the median number of harvested lymph nodes was 14 (mean 15.35), which was slightly higher than that recommended in the guideline. The maximum number of harvested lymph nodes in our data set was 55. Preoperative chemotherapy or radiotherapy tends to reduce the number of harvested lymph nodes (a mean reduction of 3.9 lymph nodes) [43].

### 4.6. Complication Rates

The overall complication rate was estimated to be 14.4% (87/605) in our study, which was comparable with the outcomes reported by single-arm studies in China, Korea, and Japan (9.9–15.5%) [28,29,30], but much lower than those of RCTs (33.1–34.8%) [34,42] (Table 5). Anastomotic leakage is a common and serious complication after rectal resection, which can lead to peritonitis, inflammation, organ failure, sepsis, or even death. The anastomotic leakage rate in our study was 3.0%, which was much lower than those of previous studies (4.1–15%) [28,29,33,36,37,42], and only Yamaguchi et al. revealed a lower anastomotic leakage rate (2.2%) than our estimates [30]. The rate of infection events in our study was 3.64%, comparable with the published studies [35,36,37,42]. Urinary retention rate (1.0%) was lower than that in previous studies (2.2–8%). Furthermore, postoperative bleeding rate (0.2%) in our study was lower than that in an RCT from Korea (0.7%) [34], but this outcome has been rarely reported in other studies. The urinary infection rate, which has been rarely reported in studies, was 2.3% in our study; Yamaguchi et al. is the only study that reported a urinary infection (theirs was 1.8%) [31]. The rate of postoperative pneumonia (0.99%) was comparable with those of previous studies (at 0–1.3%) [30,37]. The rate of ileus was 3.3%, consistent with those in previous studies (at 0–13%) [28,34,36].

### 4.7. Reoperation and Readmission

In our study, 8 of 605 patients (1.3%) were readmitted within 30 days of surgery. The median LOS due to readmission was 5 days (IQR, 2–6.25). The ACOSOG trial, which is a multicenter randomized trial conducted at 35 institutes in the United States and Canada, involved 486 patients with stage 2 or 3 rectal cancer and reported readmission rates within 30 days of 3.3% and 4.1% in laparoscopic surgery and open surgery groups, respectively [46]. In a Chinese database study, the readmission rate was 2.3% [28]. The readmission rate in our study was lower than in previous studies. The reoperation rate is an indicator that determines surgical quality and is prominently associated with long-term oncological outcomes and healthcare costs. In our study, 1.7% (10/605) of patients underwent reoperation. In the ROLARR trial, the reoperation rates of RRS and laparoscopic surgery were 3.03% and 2.74%, respectively [42]. According to US data, 5.9% of LAR patients and 8.1% of APR patients underwent reoperation [47]. In that study, reoperations after LAR were reported as predictive by a male sex (OR: 1.5), poor functional status (OR: 2.2), and operative time (OR: 1.001). As for reoperation after AR, an open approach (OR: 1.5) was one of the risk factors.

### 4.8. Recurrence and Death

Recurrence after rectal cancer surgery is not uncommon, and recurrence is more common in rectal than in colon cancer. An estimated 30–50% of patients with CRC experience recurrence or die of the cancer even after the resection. Most instances of recurrence occur within 2 years after the surgery, and the prognosis of early recurrence indicates poor survival outcomes [48]. In our study, 3.0% (18/605) of patients had local recurrence, and 66% (12/18) of the recurrence occurred within 2 years. The Japanese Society for Cancer of the Colon and Rectum guideline reported an observed local recurrence of 8.8% among patients with rectal cancer, which is much higher than our estimates [49]. A 10-year follow-up study from Singapore reported that 7.7% of patients with rectal cancer developed local recurrence, which is also higher than our estimates [50]. An RCT conducted in Korea reported that the local recurrence rates of RRS and laparoscopic surgery were 2.7% and 6.3%, respectively, which are comparable with our estimates.

The distant recurrence rate was 15.7% in our study. An RCT of RRS conducted in Korea reported systemic recurrence rates of RRS and laparoscopic surgery as 21.6% and 21.9%, respectively, which were higher than our estimates. Furthermore, a similar range was reported in other studies; a 10-year follow-up study from Singapore and a Chinese study investigating 763 patients have reported that 29.8% and 21.9% of patients with rectal cancer developed systemic recurrence after the surgery, respectively [50,51]. In terms of survival rates, the 5-year cancer-specific survival rate was 91.1% in our study. Our results indicate better patient status relative to those of other studies because the 5-year OS rate has ranged between 78–93.3% in the literature [24,33,38,39].

### 4.9. Limitations

The present study has several limitations. First, this was a retrospective study. Second, the surgical details, including ports placement, number of targets, and diverting stoma, were not collected in our data. Third, the postoperative outcomes of urinary, sexual functions, or anal functions were not analyzed. Fourth, because this study was a multi-institutional study, the perioperative and postoperative outcomes may be affected by the surgeon’s background and experience.

## 5. Conclusions

This is the first multi-institutional study in Taiwan involving 605 patients from three high-volume hospitals. Relative to participants in other studies, our participants generally had worse presurgical comorbidities, determined based on ASA and CCI scores, and better overall surgical outcomes. Crucially, the conversion rate to open surgery and anastomotic leakage rate were much lower in our studies than others in the literature. We observed no serious safety incidents despite our large sample size and long follow-up period of 8 years. Hence, based on the results of the present study, RRS is an effective and safe technique for rectal resection in high-volume hospitals.

## Figures and Tables

**Figure 1 cancers-14-04175-f001:**
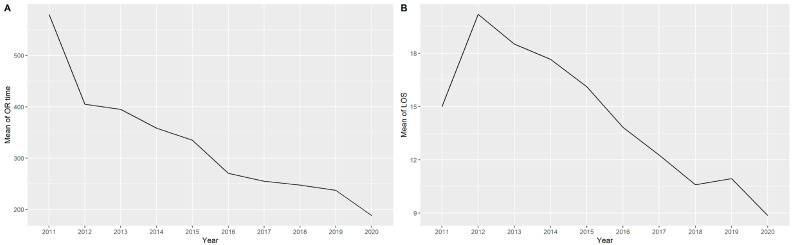
(**A**) Historical trend of operation time. (**B**) Historical trend of length of stay.

**Figure 2 cancers-14-04175-f002:**
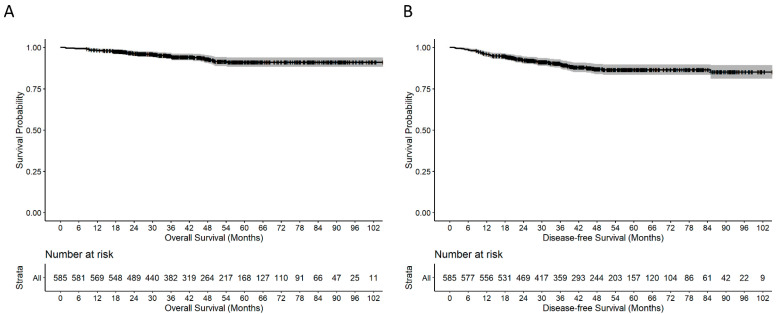
Kaplan–Meier survival curves. (**A**) Overall survival. (**B**) Disease-free survival.

**Table 1 cancers-14-04175-t001:** Demographic and baseline characteristics of 605 patients with rectal cancer undergoing robotic-assisted rectal surgery.

Characteristic	Median (IQR ^a^ or %)
Age (years, median) (range)	60 (51–67)
Gender	
Female	255 (42.1%)
Male	350 (57.9%)
Tumor distance from anal verge (cm)	
≤5 (Lower)	301 (49.7%)
6–10 (Middle)	176 (29.1%)
11–15 (Upper)	116 (19.2%)
Unknown	12 (2.0%)
AJCC Stage ^b^	
0	1 (0.2%)
I	281 (46.4%)
II	111 (18.4%)
III	194 (32.1%)
IV	13 (2.1%)
NA ^c^	5 (0.8%)
Pre-operation treatment	
CCRT ^d^	429 (70.8%)
Chemotherapy	7 (1.2%)
Radiation	18 (3.0%)
None	151 (25.0)
CCI ^e^ scores	
0, 1	536 (88.6%)
2	28 (4.6%)
≥3	41 (6.8%)
ASA ^f^ classification	
I	21 (3.5%)
II	422 (70.1%)
III	157 (26.1%)
IV	2 (0.3%)
BMI ^g^ kg/m^2^	23.7 (21.6–26.7)
Procedure	
LAR ^h^	377 (62.3%)
ISR ^i^	200 (33.1%)
APR ^j^	28 (4.6%)

^a^ IQR interquartile range; ^b^ AJCC American Joint Commission on Cancer; ^c^ NA not available; ^d^ CCRT Concurrent chemoradiotherapy; ^e^ CCI Charlson Comorbidity Index; ^f^ ASA American Society of Anesthesiologists; ^g^ BMI Body mass index; ^h^ LAR low anterior resection; ^i^ ISR, intersphenteric resection; ^j^ APR abdominoperineal resection.

**Table 2 cancers-14-04175-t002:** Intraoperative Safety and Clinical Outcomes of 605 patients with rectal cancer undergoing robotic-assisted rectal surgery.

Intraoperative Safety	
Characteristic	Median (IQR ^a^ or %)
Conversions to open surgery	1 (0.2%)
Console Time (min, median) (range)	211 (172–256)
Operation Time (min, median) (range)	270 (210–335)
Estimated blood loss (mL, Median)	50 (30–100)
Blood transfusion during surgery	10 (1.7%)
Rehospitalization within the 30-day postoperative period	8 (1.3%)
Reoperation within the 30-day postoperative period	10 (1.7%)
Death during surgery	0 (0.0%)
Death within the 30-day postoperative period	0 (0.0%)
Pathological outcomes and Oncological outcomes	
Characteristic	Median (IQR ^a^ or %)
Harvested Lymph Node	14 (10–20)
Distal resection margin	
Free	584 (96.5%)
Positive	21 (3.5%)
Circumferential resection margin	
Free	575 (95.0%)
Positive	30 (5.0%)
Relapse	113 (18.7%)
Local recurrence	18 (3.0%)
Distant metastasis	95 (15.7)
Cancer-specific death during follow-up period	39 (6.4%)

^a^ IQR interquartile range.

**Table 3 cancers-14-04175-t003:** Postoperative complications of 605 patients with rectal cancer undergoing robotic-assisted rectal surgery.

Complications	Number (%)
Post-operative bleeding	1 (0.2%)
Anastomosis leakage	18 (3.0%)
Ileus	20 (3.3%)
Infection events ^a^	22 (3.6%)
Urinary retention	6 (1.0%)
Urinary infection	14 (2.3%)
Pulmonary complication	6 (1.0%)
Total	87 (14.4%)

^a^ Infection events included intraabdominal infection, intraabdominal abscess, and surgical site infection.

**Table 4 cancers-14-04175-t004:** Surgery characteristics of studies from relevant literatures.

Author (Year, Design)	Country	Patient Number	Surgery Type	Cancer Stage	OR Time (Minutes)	Lymph Node Yields	LOS (Days)	Conversion
Present study	Taiwan	R: 605	LAR, APR, ISR	I, II, III, IV	284.11	15.35	13.5	0.17%
Katsuno [30] (2020, Cohort)	Japan	R: 115	LAR, APR, ISR	I, II	341	NA	11 *	0
Yamaguchi [31] (2018, Cohort)	Japan	R: 551	HAR, LAR, ISR, APR, Hartmann	I, II, III, IV	257	NA	7 *	0
Kim [34] (2016, Cohort)	Korea	R: 60	LAR, APR	I, II, III, IV	466.8	20.1	8.6	0%
Tang [36] (2016, Cohort)	China	R: 392	LAR, APR, Hartmann	I, II, III, IV	297	14.6	12.1	1.80%
Lim [37] (2017, Cohort)	Korea	R: 74	LAR, ISR, CAA, APR	CR, I, II, III	365.2	11.6	NA	1.40%
		L: 64			311.6	14.7	NA	6.30%
Chen [11] (2020, Cohort)	Taiwan	R: 88	TME	CR, I, II, III	NA	NA	NA	NA
		L: 37			NA	NA	NA	NA
		O: 175			NA	NA	NA	NA
Huang [24] (2017, Cohort)	Taiwan	R: 40	LAR, ISR	I, II, III	274.4	NA	12.9	NA
		L: 38			235.4	NA	11.7	NA
Somashekhar [38] (2015, RCT)	India	R: 25	LAR, AR	NA	R: 310.3	16.88	7.52	NA
		O: 25			L: 246.9	15.2	13.24	NA
Jayne [29] (2017, RCT) ROLARR	Multinational (Ten countries)	R: 237	LAR, APR, HAR, Hartmann (High anterior resection)	I, II, III, IV	R: 298.5	24.1	8.2	8.10%
		L: 234			L: 261	23.2	8	12.20%
Kim [35] (2018, RCT)	Korea	R: 66	LAR, APR, Hartmann	I, II, III, IV	R: 339.2	18	10.3	1.50%
		L: 73			L: 227.8	15	10.8	0%
Sujatha-Bhaskar [39] (2017, Database)	United States	R: 905	APR, Proctectomy (incl. LAR)	I, II, III	NA	15.7	NA	7%
		O: 3399			NA	14.8	NA	NA
		L: 2009			NA	15.2	NA	14%
Hyde [32] (2019, Database)	United States	R: 6035	LAR	I, II, III, IV	NA	17	6.3	7.45
		O: 21,421			NA	16.4	7.8	NA
		L: 13,826			NA	16.8	6.8	14.95
Chang [28] (2020, Database)	China	R: 1145	APR, LAR, APR, Hartmann	Benign, I, II, III, IV	NA	17	NA	NA
Author (year, design)	Country	Patient number	Reoperation	Transfusion	Blood loss (mL)		Positive CRM	Recurrence
Present study	Taiwan	R: 605	1.70%	1.65%	72.58		4.96%	Local: 2.96% Systemic: 15.67%
Katsuno [30] (2020, Cohort)	Japan	R: 115	NA	0	20		NA	Local: 3.5% Systemic: 20.0%
Yamaguchi [31] (2018, Cohort)	Japan	R: 551	NA	0	10		NA	NA
Kim [34] (2016, Cohort)	Korea	R: 60	NA	NA	74.2		11.70%	Local: 1.9% Systemic: 26.4%
Tang [36] (2016, Cohort)	China	R:392	1.8%	NA	67.5		2.30%	Local: 2.3%
Lim [37] (2017, Cohort)	Korea	R: 74	NA	NA	NA		NA	Local: 2.7% Systemic: 18.9%
		L: 64	NA	NA	NA		NA	Local: 6.3 Systemic: 15.6%
Chen [11] (2020, Cohort)	Taiwan	R: 88	NA	NA	NA		3.40%	Local: 2.30% Systemic: 21,6%
		L: 37	NA	NA	NA		16.20%	Local: 21.60% Systemic: 35.1%
		O: 175	NA	NA	NA		12%	Local:6.90% Systemic: 20.6%
Huang [24] (2017, Cohort)	Taiwan	R: 40	NA	NA	41.9		NA	NA
		L: 38	NA	NA	55.1		NA	NA
Somashekhar [38] (2015, RCT)	India	R: 25	NA	NA	165.14		0%	NA
		O: 25	NA	NA	406.04		0%	NA
Jayne [29] (2017, RCT) ROLARR	Multinational (Ten countries)	R: 237	NA	NA	NA		5.10%	NA
		L: 234	NA	NA	NA		6.30%	NA
Kim [35] (2018, RCT)	Korea	R: 66	3.03%	NA	100		6.10%	NA
		L: 73	2.74%	NA	50		5.50%	NA
Sujatha-Bhaskar [39] (2017, Database)	United States	R: 905	NA	NA	NA		4.75%	NA
		O: 3399	NA	NA	NA		7.62%	NA
		L: 2009	NA	NA	NA		4.87%	NA
Hyde [32] (2019, Database)	United States	R: 6035	NA	NA	NA		NA	NA
		O: 21,421	NA	NA	NA		NA	NA
		L: 13,826	NA	NA	NA		NA	NA
Chang [28] (2020, Database)	China	R: 1145	0.80%	NA	NA		1.30%	NA
Author (year, design)	Country	Patient Number	30 Day Readmission		30 Day Mortality		Disease Free Survival (DFS)	
Present study	Taiwan	R: 605	1.32%		0%		5y: 86.3%	
Katsuno [30] (2020, Cohort)	Japan	R: 115	NA		NA		I: 93.5% II: 100% III: 83.8%	
Yamaguchi [31] (2018, Cohort)	Japan	R: 551	NA		NA		I: 93.6% II: 75% III: 77.6%	
Kim [34] (2016, Cohort)	Korea	R: 60	NA		NA		4y: 72.8%	
Tang [36] (2016, Cohort)	China	R: 392	NA		0.5%		3y: 74.3%	
Lim [37] (2017, Cohort)	Korea	R: 74	NA		NA		NA	
		L: 64	NA		NA		NA	
Chen [11] (2020, Cohort)	Taiwan	R: 88	NA		NA		NA	
		L: 37	NA		NA		NA	
		O: 175	NA		NA		NA	
Huang [24] (2017, Cohort)	Taiwan	R: 40	NA		NA		NA	
		L: 38	NA		NA		NA	
Somashekhar [38] (2015, RCT)	India	R: 25	NA		NA		NA	
		O: 25	NA		NA		NA	
Jayne [29] (2017, RCT) ROLARR	Multinational (Ten countries)	R: 237	NA		0.80%		NA	
		L: 234	NA		0.90%		NA	
Kim [35] (2018, RCT)	Korea	R: 66	NA		NA		NA	
		L: 73	NA		NA		NA	
Sujatha-Bhaskar [39] (2017, Database)	United States	R: 905	NA		0%		NA	
		O: 3399	NA		0%		NA	
		L: 2009	NA		0.16%		NA	
Hyde [32] (2019, Database)	United States	R: 6035	NA		0.9		NA	
		O: 21,421	NA		1.1		NA	
		L: 13,826	NA		1.5		NA	
Chang [28] (2020, Database)	China	R: 1145	2.30%		0.10%		NA	

R: Robot Assisted Surgery, L: Laparoscopic surgery, O: Open surgery, LAR: Low Anterior Resection, APR: Abdominoperineal resection, ISR: intersphincteric Resection, TME: Total Mesorectal Excision, HAR: Higher Anterior Resection, CAA: Coloanal anastomosis, NA not available. * Median.

**Table 5 cancers-14-04175-t005:** Complication characteristics of studies from relevant literature.

Author (Year, Design)	Country	Patient Number	Overall Complication Rate	Anastomotic Leakage	Incisional Hernia	Surgical Site Infection	Ileus
Present study	Taiwan	R: 605	13.39%	2.98%	0%	3.64%	3.31%
Katsuno [30] (2020, Cohort)	Japan	R: 115	14.80%	6.10%	NA	1.70%	NA
Yamaguchi [31] (2018, Cohort)	Japan	R: 551	15.50%	2.20%	NA	NA	NA
Kim [34] (2016, Cohort)	Korea	R: 60	15%	5%	NA	NA	3%
Tang [36] (2016, Cohort)	China	R:392	9.9%	4.10%	NA	NA	NA
Huang [24] (2017, Cohort)	Taiwan	R: 40	15.00%	7.50%	NA	NA	0.00%
		L: 38	18%	5%	NA	NA	13%
Somashekhar [38] (2015, RCT)	India	R: 25	0.00%	NA	NA	NA	NA
		O: 25	20.00%	NA	NA	NA	NA
Jayne [29] (2017, RCT) ROLARR	Multinational (Ten countries)	R: 237	33%	15%	NA	9%	NA
		L: 234	31.70%	17.40%	NA	8.30%	NA
Kim [35] (2018, RCT)	Korea	R: 66	34.80%	12.10%	NA	NA	9.10%
		L: 73	23%	7%	NA	NA	12%
Sujatha-Bhaskar [39] (2017, Database)	United States	R: 905	NA	NA	NA	NA	NA
		O: 3399	NA	NA	NA	NA	NA
		L: 2009	NA	NA	NA	NA	NA
Chang [28] (2020, Database)	China	R: 1145	16.30%	4.20%	NA	NA	1.30%
Author (year, design)	Country	Patient number	Abdominal bleeding	Urinary retention	Urinary infection	Pneumonia	Fecal incontinence
Present study	Taiwan	R: 605	0.17%	0.99%	2.31%	0.99%	0.17%
Katsuno [30] (2020, Cohort)	Japan	R: 115	NA	3.50%	NA	NA	NA
Yamaguchi [31] (2018, Cohort)	Japan	R: 551	NA	2.20%	1.80%	1.30%	NA
Kim [34] (2016, Cohort)	Korea	R: 60	NA	NA	NA	NA	NA
Tang [36] (2016, Cohort)	China	R:392	NA	NA	NA	0.00%	NA
Huang [24] (2017, Cohort)	Taiwan	R: 40	NA	NA	NA	NA	NA
		L: 38	NA	NA	NA	NA	NA
Somashekhar [38] (2015, RCT)	India	R: 25	NA	8.00%	NA	NA	NA
		O: 25	NA	20.00%	NA	NA	NA
Jayne [29] (2017, RCT) ROLARR	Multinational (Ten countries)	R: 237	NA	NA	NA	NA	NA
		L: 234	NA	NA	NA	NA	NA
Kim [35] (2018, RCT)	Korea	R: 66	0.70%	NA	NA	NA	NA
		L: 73	0%	NA	NA	NA	NA
Sujatha-Bhaskar [39] (2017, Database)		L: 5935	NA	NA	NA	NA	NA
	United States	R: 905	NA	NA	NA	NA	NA
		O: 3399	NA	NA	NA	NA	NA
Chang [28] (2020, Database)		L: 13,826	NA	NA	NA	NA	NA
	China	R: 1145	NA	2.50%	NA	NA	NA

R: Robot Assisted Surgery, L: Laparoscopic surgery, O: Open surgery, NA not available.

## Data Availability

The data presented in this study are available in this article.

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
