# Peer review of "Clinical Safety and Effectiveness of Robotic-Assisted Surgery in Patients with Rectal Cancer: Real-World Experience over 8 Years of Multiple Institutions with High-Volume Robotic-Assisted Surgery"

_cancers, 2022, doi:10.3390/cancers14174175_

Round 1

Reviewer 1 Report

This is a well-conducted and well-written paper and I have no major criticism.

However, I don't quite commend the statement from the authors: "The overall surgical and oncological outcomes were equivalent or superior to those estimated in other studies." I think this conclusion is just in the Taiwanese context, let alone the present comparison is not based on comprehensive review from other reports domestically and even from Asian countries. I suggest inclusion of the following reports with some discussion to make their statement more convincing before the final acceptance for publication: 

1. An et. al. Asian Journal of Surgery 44 (2021) 199-205

2.  Feng et. al. Asian Journal of Surgery 44 (2021) 440-451

3 . Tong et. al. Asian Journal of Surgery 44 (2021) 1549

4  . Han et. al. Asian Journal of Surgery 43 (2020) 880-890

5. Chen et. al. J Formos Med Assoc.  2022 Aug;121(8):1532-1540

Author Response

Thank you for your valuable comments on our manuscript. We have included the above reports in the references (Ref 13, 14, and 15).

Reviewer 2 Report

This is an interesting study that highlighted a promising role for robotics in rectal cancer surgery and their findings are consistent with emerging evidence across the globe on this topic. The authors have demonstrated equivalent or superior surgical outcomes when robotic surgery is used effectively at high volume centres. Their conversion rates are very low but it must be highlighted that the median BMI reported in this study is generally much lower than the Western population where the overweight or obese narrow male pelvis often pose significant operative challenges even in robotic assisted surgery.

The authors have reported in Table 2 that their 30-day reoperation rate is 2.5% (15 patients). However in Section 3.4 (Postoperative Complications), the authors have stated that all post-operative complications (14.4%) are of Grade 1 Clavien Dindo classification and that the patients recovered after conservative treatment. These two statements are inconsistent as a 30-day post-operative return to theatre is classified as a Grade 3 Clavien Dindo complication. The authors will need to clarify this and the reasons for those that needed reoperations.

It would also be useful to know the number of robotic surgeons involved from these centres, their robotic experiences and their respective case numbers contributed to this study. This will enable to readership to appreciate the breadth and any variability of robotic experiences across these centres that are required to generate these outcomes.  

Kindly remove paragraph lines from 87 to 101; and from 243 to 246, as it is not needed.

Overall, the authors have made an excellent effort to collate these data and generate meaningful outcomes that are highly pertinent to future of robotic surgery in rectal cancer surgery. However I would appreciate it if some of the comments above are addressed or clarified. Many thanks.

Author Response

Thank you for giving us the opportunity to resubmit our manuscript. We have revised the manuscript substantially according to the reviewers’ comments and have responded to those comments on a point-by-point basis as follows. The revised portions of the manuscript are shown in red color in the manuscript file. In addition, we have also amended the manuscript according to journal requirement that you mentioned in previous decision letter. Hopefully our revised manuscript will elucidate all the unclear points and make publication in the Cancers feasible.

Reviewer 3 Report

Very good results (few conversion, very few AL rate , low rate of positive CRM) for a 50% low RC series

3 hospitals but which number of surgeons ?

No surgical details on the procedure (ports placement, number of targets) / Diverting stoma ???

Learning curve must be emphasized

Low conversion rate : due to low median BMI (not discussed)

Some table with a lot of NA results, some series should be delete

References with old publications

Author Response

(The authors gave the same response as above.)

Round 2

Reviewer 3 Report

no comments